# Electrocatalyst Derived from NiCu–MOF Arrays on Graphene Oxide Modified Carbon Cloth for Water Splitting

Lisha Jia, Pawel Wagner *[ID] and Jun Chen *[ID]

ARC Centre of Excellence for Electromaterials Science, Intelligent Polymer Research Institute,
Australian Institute for Innovative Materials, Innovation Campus, University of Wollongong, Squires Way,
North Wollongong, NSW 2500, Australia; lj027@uowmail.edu.au
* Correspondence: pawel@uow.edu.au (P.W.); junc@uow.edu.au (J.C.)

**Abstract:** Electrocatalysts are capable of transforming water into hydrogen, oxygen, and therefore into energy, in an environmentally friendly and sustainable manner. However, the limitations in the research of high performance catalysts act as an obstructer in the development of using water as green energy. Here, we report on a delicate method to prepare novel bimetallic metal organic framework derived electrocatalysts (C–NiCu–BDC–GO–CC) using graphene oxide (GO) modified carbon cloth as a 3D flexible and conductive substrate. The resultant electrocatalyst, C–NiCu–BDC–GO–CC, exhibited very low electron transfer resistance, which benefited from its extremely thin 3D sponge-like morphology. Furthermore, it showed excellent oxygen evolution reaction (OER) activity, achieving 10 mA/cm$^2$ at a low overpotential of 390 mV in 1 M KOH electrolyte with a remarkable durability of 10 h.

**Keywords:** MOF arrays; graphene oxide; carbon cloth; water splitting

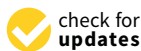



## 1. Introduction

The start of 2022 has been marked by a global energy crisis [1]. Many researchers believe that as the source of life on Earth, water maintains enormous power [2]. Hence, most countries have endeavored to explore the power derived from water and intend to regard water as a green and sustainable fuel to replace traditional fossil energy. For example, hydropower can be employed to generate electricity and a variety of turbines have been studied and can now be commercialized on a large scale for great output hydropower [3]. In fact, the present civilization on Earth is still on the way to discovering how to unitize water. Electrochemical water splitting is one of most popular methods and has gained the attention of more and more researchers to reproduce $H_2$ as a sustainable and clean energy of the 21st century. $H_2$ is a promising green fuel with no carbon product emissions, which is one of the main reasons for climate change. Specifically, 1 g $H_2$ can produce 140 kJ after full combustion in air at room temperature, which is about three times that of 1 g of gasoline [4]. Water splitting basically contains two key electrode reactions: hydrogen evolution reaction (HER) at a cathode and oxygen evolution reaction (OER) at an anode. However, due to the limited efficiency or exorbitant cost of electrocatalysts, the development of splitting water into $H_2$ and $O_2$ has been hindered. By far, noble metal platinum (Pt) is the most active element for HER and noble metal oxides ($IrO_x$ and $RuO_x$) are the benchmark electrocatalysts for OER [5,6]. Hence, designing and fabricating an earth-abundant electrocatalyst, which is capable of lowering the energy barrier HER and OER is of great importance [7].

Metal organic frameworks (MOFs) are organic–inorganic hybrids assembled from metal ions (or clusters) and organic ligands [8,9]. MOFs have been widely employed as catalysts, precursors, or substrates in electrocatalysis, benefiting from MOFs' unique porous structure, large surface area, and more exposed active sites, etc. [10–12]. However, it is difficult to use them directly as catalysts, because of their undesirable conductivity and

instability in an aqueous system [13–15]. To improve their conductivity and catalytic performance, MOF derived materials have recently been investigated and compared with their initial MOFs, since MOFs were first used as templates for the synthesis of porous carbons, as reported by Liu et al. in 2008 [16–20]. Among these strategies, metal/metal oxide nanoparticles exhibit promising catalytic performance and are widely used as electrode materials in electrocatalysis and can be obtained by carbonizing MOFs at different temperature gradients [21–24]. Additionally, GO can be utilized as the secondary conductive support through hybridization with MOFs to improve the MOFs' conductivity [25]. Meanwhile, developing bimetallic MOFs is also a common approach to enhance the catalytic performance [26]. In this regard, nickel is beneficial from its earth abundance, corrosion resistance, and thermal stability, and is frequently applied in a wide variety of fields (batteries, super capacitors, and catalysis, etc.). Furthermore, Ni-based MOF derivatives are continually employed in the process of water electrocatalysis [27–29]. Theoretically, Cu presents a high electrical conductivity, which is advantageous to accelerate charge transfer and further enhance the catalytic activity [30,31]. However, up to now, NiCu–MOFs and their derivatives have not been fully explored yet.

In order to fabricate a working electrode using powder catalysts, generally, catalyst ink needs to be prepared using black carbon to ameliorate the MOF powder's poor conductivity and nafion (5 wt%) as a binder to combine all compounds together. The time-consuming procedure of preparing catalyst ink may have an effect on the activity of the original powdered MOFs [32]. Furthermore, a mechanical stability issue may be caused when the ink is coated on the electrode (glassy carbon, F-doped tin oxide, and nickel foam, etc.) due to the high pressure of gas generated inside the pores of the catalytic material [33].

Herein, we proposed a delicate procedure to synthesize a bimetallic MOF derived electrocatalyst, carbonized NiCu–BDC on GO modified carbon cloth. The catalysts remained the beneficial sponge-like structure of the NiCu-layered double hydroxide arrays (NiCu–LDH) by in situ transformation. The structure can increase the mass transport between the surface of the catalysts and electrolyte and make the electron transfer faster compared with the powdered catalysts. Moreover, arrays directly grown on the surface of GO/CC could effectively avoid the rupture of the catalyst coating, besides, the close and strong binding could thoroughly eliminate the effect from the carbon black and nafion binder. As a result, the mechanical stability of catalysts was significantly improved. Furthermore, the 3D carbon cloth can work as a self-supported and conductively flexible substrate. Finally, the carbonization process has a great value in improving the conductivity of the original MOFs, maintaining the morphology of the MOF arrays. Based on the above theoretical knowledge, the performance of obtained bimetallic C–NiCu–BDC/GO/CC on HER, OER, and overall water splitting were analyzed.

## 2. Experimental Section

### 2.1. Chemical and Materials

Copper(II) nitrate trihydrate ($Cu(NO_3)_2 \cdot 3H_2O$), >99.9%, nickel(II) nitrate hexahydrate ($Ni(NO_3)_2 \cdot 6H_2O$), 99.9%, and terephthalic acid (BDC) 98% were purchased from Sigma-Aldrich. Hexadeyltrimethylammonium bromide (CTAB) >98% and potassium hydroxide (KOH) were purchased from Chem-Supply. *N,N*-dimethylformamide (DMF) and methanol were purchased from RCL Labscan. All chemical reagents were used without any further purification. Carbon cloth was ordered from FuelCellStore. GO was synthesized by the hammer method [34]

### 2.2. Preparation of Electrocatalysts

2.2.1. Preparation of GO Modified CC

Carbon cloth was sonicated with acetone, ethanol, and DI water for 30 min, in sequence, then immersed in the 3M $HNO_3$ solution overnight for further cleaning and then washed with DI water to remove the acid. The CC was ready to be used after being dried using flowing Ar. For the GO deposition step, a 1 cm × 1 cm platinum sheet and 1 cm × 1 cm CC

functioned as the counter and working electrode, respectively, using 1 mg/mL GO solution as the electrolyte, applying 1 mA/cm$^2$ for 15 min on an anode electrode, before finally hanging the GO/CC and drying it naturally at room temperature.

### 2.2.2. Deposition of NiCu–LDH on GO/CC

NiCu–LDH was prepared by the hydrothermal method, where 25 mM Cu(NO$_3$)$_2$·3H$_2$O, 17 mM Ni(NO$_3$)$_2$·6H$_2$O, and 500 mg CTAB were dissolved in 30 mL methanol and 6 mL DI water, then the mixture was placed into a 60 mL stainless steel autoclave, two pieces of GO/CC was placed into the mixture solution and the autoclave sealed tightly, before the vessel was heated at 200 °C for 24 h. After cooling down to room temperature and rinsing the GO/CC with DI water three times and then drying in an oven at 60 °C for 2 h, finally, a purple NiCu–LDH was deposited on GO/CC, NiCu–LDH/GO/CC arrays were obtained.

### 2.2.3. Deposition NiCu–BDC on GO/CC

The obtained precursor NiCu–LDH/GO/CC was in situ transformed into MOFs by adding one piece of precursor into a glass vial with reaction solution (20 mg BDC dissolved in 3.64 mL DMF and 0.36 mL DI water). The vial was placed in an oven at 100 °C for 24 h, after cooling down naturally. The piece of CC was washed with DMF and ethanol three times, respectively. Finally, it was dried at 60 °C in the oven and the MOF arrays were ready for the carbonization step.

### 2.2.4. Carbonization of NiCu–BDC/GO/CC

NiCu–BDC/GO/CC MOF arrays were placed in a high temperature ceramic covered with a refractory glass and put into a furnace to carbonize the MOFs at 450 °C for 4 h in an Ar atmosphere. After cooling to ambient temperature, the C–NiCu–BDC/GO/CC was obtained.

### 2.2.5. Preparation of C–NiCu–BDC/GO/CC Powder

As a comparison, the C–NiCu–BDC/GO/CC powder was prepared by adding GO solution, instead of GO/CC, directly into the reaction solution described in Section 2.2.2 and following the same synthesis procedure as the C–NiCu–BDC/GO/CC arrays. After carbonization, C–NiCu–BDC/GO powder was obtained and the loading mass was 0.8 mg/cm$^2$.

### 2.3. Material Characterization

The powder X-ray diffraction (XRD) of pyrolyzed MOF arrays on CC and its powder counterpart were measured on a PANalytical Empyrean with a Cu K radiation source at a generator voltage of 40 KV and a generator current of 40 mA with a scanning speed of 3°/min. Raman spectroscopy was conducted on a HORIBA scientific-LabRAM HR Evolution with an excitation wavelength of 563 nm. A scanning electron microscope (JEOL JSEM-6490 LV) at an operating voltage of 15 KV and a scanning transmission electron microscope (AMF 200) was used to obtain SEM and STEM images. X-ray photoelectron spectroscopy (XPS) was performed on a Thermo Fisher Scientific NEXSA Surface Analysis system. Thermogravimetric analysis (TGA) was measured on a NETZSCH TG 209 in a N$_2$ atmosphere with a heating rate of 3°/min.

### 2.4. Electrochemical Measurements

A three-electrode system with two compartments was used to measure the HER and OER electrocatalysis process. To be specific, a 1 cm × 1 cm platinum sheet and Hg/HgO were used as the counter electrode and reference electrode, respectively. To fabricate the working electrode, 1 cm × 1 cm C–NiCu–BDC/GO/CC was directly used as the working electrode with a loading mass of 0.8 mg/cm$^2$. For comparison, catalyst ink of C–NiCu–BDC/GO powder was prepared with 0.8 mg/cm$^2$, and 1 cm × 1 cm of clean carbon cloth with 1 M KOH was also used as the electrolyte. All electrochemical potential values for the

three-electrode configuration used in this study were calculated from E (Hg/HgO) to E (RHE) via the following equation:

$$E\ (RHE) = \ E\left(\frac{Hg}{HgO}\right) + 0.098 + 0.05916 \times \ pH$$

where E (Hg/HgO) is the measured potential against the reference electrode.

The overall water splitting performance was carried out in a two-electrode system. Both of the working electrodes were C–NiCu–BDC/GO/CC and freshly prepared prior to each experiment. All measurements were carried out on a CHI660 potentiostat using the automatic iR compensation function and conducted in 1 M KOH with Ar bubbled.

Electrochemical impedance spectroscopy (EIS) was conducted at the potentials at which the current density was 10 mA/cm$^2$ and at constant current density of 10 mA/cm$^2$. The frequency range for all EIS testing was from 0.01 Hz to 100 kHz. The durability measurements of the catalysts were carried out with i–t curve at the potential of current density at 10 mA/cm$^2$ and v–t curves at a constant current density of 10 mA/cm$^2$.

## 3. Results and Discussion

### 3.1. Structural Characterization

The bimetallic C–NiCu–BDC/GO/CC catalysts were synthesized through a four-step growth procedure, as shown in Figure 1a. The optimal temperature for the NiCu–LDH arrays was investigated at 180, 200, and 220 °C; the results in Figure S1 show that the morphology of LDH was uniform and neat only at 200 °C. After the transformation of the MOFs, NiCu–BDC–GO arrays were obtained and CV curves measured to identify their stability in 0.1 M Na$_2$SO$_4$. Figure S2 indicates the poor mechanical and chemistry stabilities of pristine MOFs. Through the carbonization of sponge-like NiCu-BDC/GO/CC MOFs, C–MOFs maintained the valuable morphology of the precursor, in order to improve the stability compared to the MOF arrays before carbonization, which makes C–MOFs arrays more suitable to act as a promising catalyst in an aqueous system. As shown in Figure 1b,c, it can be clearly observed that the extremely thin GO layer that was deposited on the surface of bare CC and GO was extremely crucial to function as anchors to hook the deposited NiCu–LDH precursor (Figure 1c). The SEM image (Figure 1e) showed that the sponge-like arrays of NiCu-BDC/GO/CC were uniformly deposited on GO/CC after in situ transformation from the precursor. In the final step, the pyrolysis temperature was determined at 450 °C based on the TGA curves of the NiCu–MOF arrays (Figure S3). Carbonized NiCu–BDC/GO/CC (Figure 1f) was expected to possess desirable mechanical and chemical stability in the aqueous electrolyte while retaining the same morphology as the MOFs.

The XRD pattern (Figure 2a) of powdered MOFs (blue line) exhibited high crystalline MOF structure, typical reflections of (200) and (110) belonged to Ni–BDC and Cu–BDC, respectively. The MOF arrays (green line) also obtained peaks at 8° and 10° for planes (200) and (110), which were formed in the in situ MOF transformation step. In the same process, a distinct NiCu–LDH peak (003) at 12 disappeared, as shown in Figure S4. However, compared to the powdered MOFs, the peaks of powder between 12° to 35° could not be observed in the pattern of arrays due to the high background of reflections of bare CC. After carbonization, all peaks of the MOF arrays in the XRD pattern disappeared (orange line).

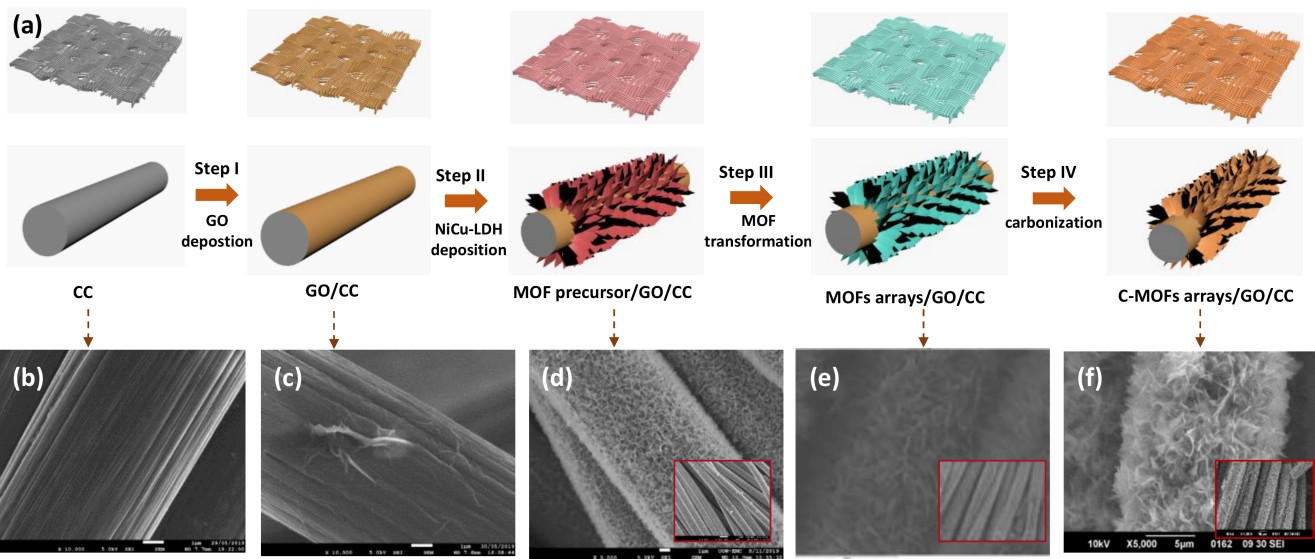

**Figure 1.** (**a**) The schematic illustration of synthesizing C–NiCu–MOFs; (**b**,**c**) SEM images of CC with pre-treatment and GO/CC; (**d**–**f**) SEM images of NiCu–LDH/GO/CC, NiCu–BDC/GO/CC, and C–NiCu–BDC/GO/CC.

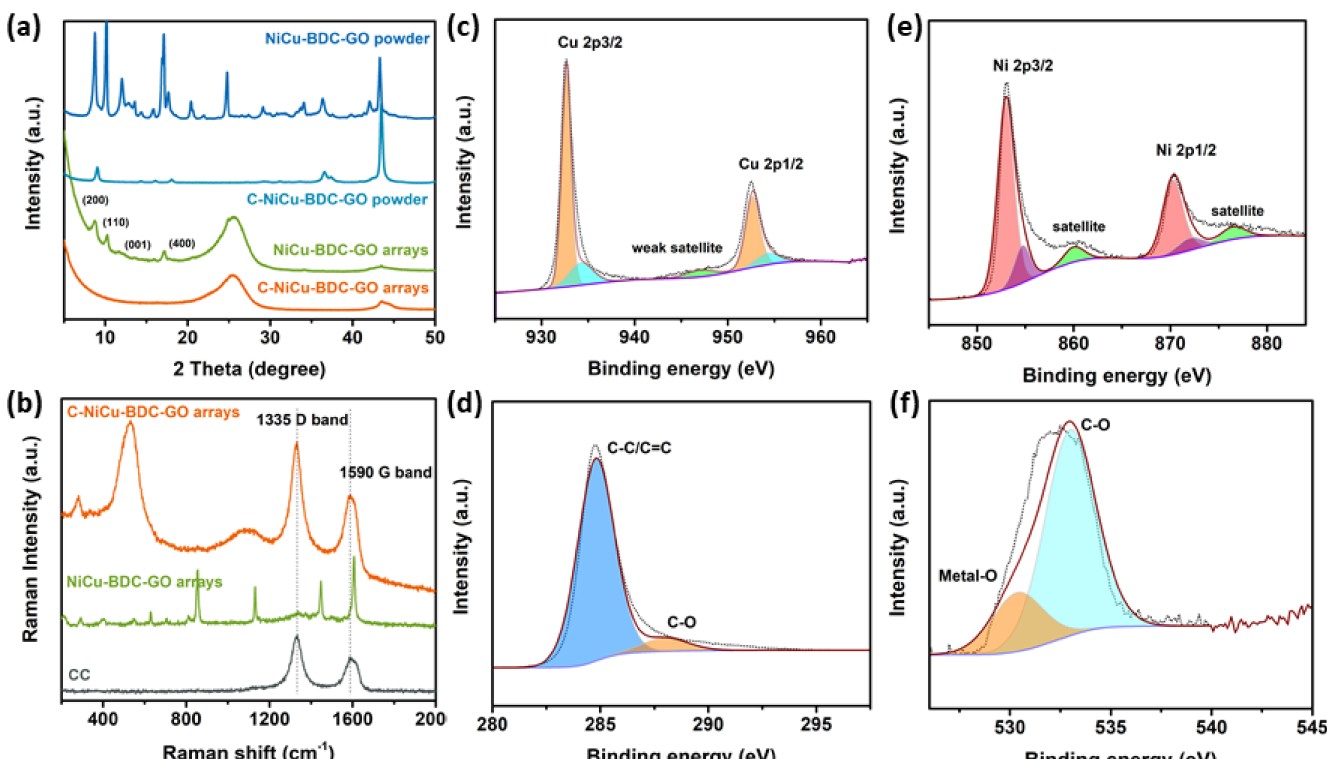

**Figure 2.** (**a**) XRD patterns of the as-prepared catalyst C–NiCu–MOF arrays and the comparison with C–NiCu–MOF powder; (**b**) Raman spectra of C–NiCu–MOF arrays and powder; (**c**) the fitted peaks corresponded to Cu⁰ and Cu⁺ 2p3/2 (932.6 eV) and 2p1/2 (952.7 eV); (**d**) the fitted peaks corresponded to graphitic carbon (284.8 eV); (**e**) the fitted peaks related to $Ni^{2+}$ 2p3/2 (852.9 eV) and 2p1/2 (870.4 eV); (**f**) O 1s (531.1 and 532.9 eV).

The surface chemical compositions of the C–MOFs and valence information of Cu and Ni in the C–MOFs arrays were confirmed by X-ray photoelectron spectroscopy (XPS). The survey scan on the C–MOFs arrays demonstrated the existence of C, O, Ni, and Cu elements. Furthermore, Figure 2b shows the XPS high-resolution spectrum of Cu2p, where

the peaks at 932.6 eV (2p3/2) and 952.7 eV (2p1/2) demonstrate the presence of $Cu^0$ or $Cu^+$, and the weak satellite can be attributed to the $Cu^+$. As shown in Figure 2c, the peaks at 852.9 and 870.4 eV represent the characteristic Ni2p3/2 and Ni2p1/2 and two satellites suggest the electronic state of Ni is $Ni^{2+}$. Furthermore, the C 1s spectrum showed graphitic carbon at 284.8 eV and C–O at 287 eV (Figure 2d), and the density of the O 1s spectrum of the C–MOFs arrays (Figure 2e) was extremely weaker than that of the MOF arrays shown in Figure S5, due to the process of carbonization in an Ar atmosphere.

In addition, in the Raman spectrum, there were two obvious peaks located around 1335 and 1590 $cm^{-1}$, which can be attributed to the D and G bands of graphite carbon, respectively. The D band originates from graphite defects, while the G band corresponds to the $sp^{2-}$ bonded hexagonal lattice of graphite. Furthermore, the G band of C–NiCu–MOFs arrays was significantly stronger than that of CC. This indicates that there are more functional groups connected to the carbon in the C–NiCu–MOF arrays compared to bare CC. Moreover, the peaks in the Raman spectrum of the NiCu–MOF arrays proved the existence of Ni–BDC and Cu–BDC and also reconfirmed that the MOFs synthesized in this work were bimetallic NiCu–BDC/GO/CC.

Scanning transmission electron microscopy (STEM) confirmed the $Cu^0$ and $Cu_2O$ in the as-prepared C–NiCuMOFs. Figure 3a,b point out the nanoparticles with diameters around 10 nm and the carbonized MOFs were comprised of Cu, $Cu_2O$, and NiO components. The energy dispersive spectroscopy (EDS) elemental mapping analysis showed the uniform distribution of elemental Cu, C, and O (Figure 3d,e,g,h) and reconfirmed the results of the catalyst's composition in the XPS spectrum. Additionally, the atom percentage of two metal elements (Ni and Cu) were determined by the EDS analysis, where the Cu element was around 20% and Ni was only 3%, as shown in Figure S6, which suggests that the Cu and $Cu_2O$ nanoparticles were the catalytic domain of the C–MOF arrays for the following electrochemical measurements.

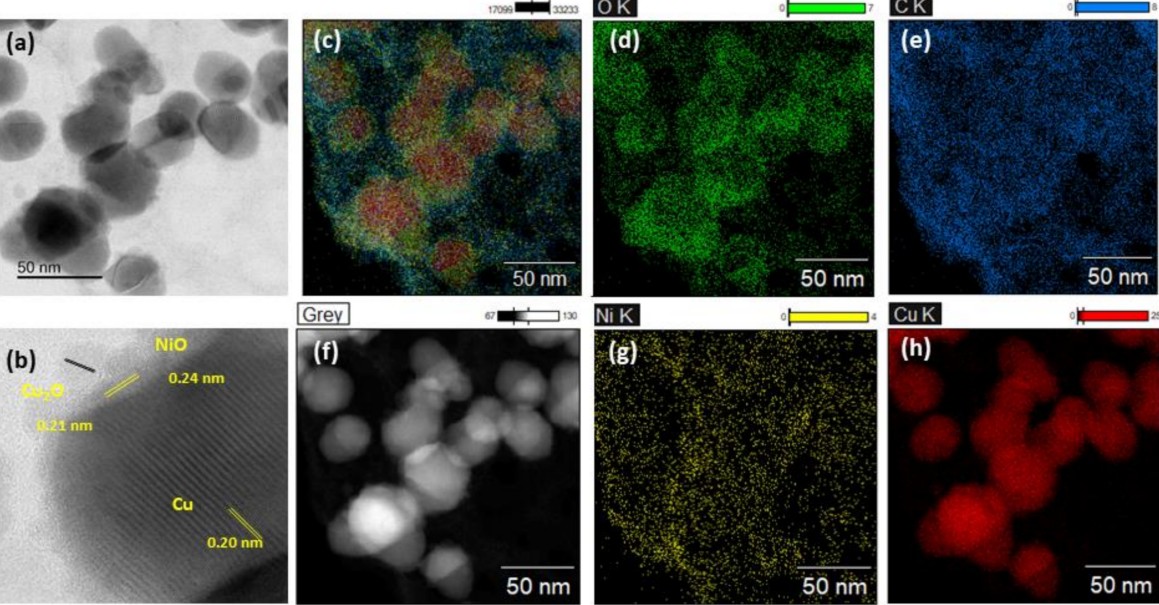

**Figure 3.** (**a,b,f**) HAADF–STEM images with different resolutions. (**c,d,e,g,h**) The corresponding elemental mapping results of the C–NiCu–MOF arrays.

### 3.2. Electrochemical Measurements

As shown, the iR-corrected polarization curves (Figure 4a) were tested in 1 M KOH using the three-electrode system. For comparison with other catalysts, many of the tested catalytic materials are currently benchmarked at rather low current densities (often between 10 and 100 $mA/cm^2$) [35]. In this work, the current density was tested at 10 $mA/cm^2$. For OER polarization curves, the C–NiCu–MOF array had the lowest overpotential of 390 mV

at 10 mA/cm$^2$, which was much lower than that of the C–NiCu–MOF powder (510 mV) and almost comparable to the commercial $RuO_2$ electrocatalyst (320 mV). For the HER polarization curves, the overpotential of the two catalysts did not show much difference (400 mV), and was still much smaller than that of bare CC (750 mV). Furthermore, the C–NiCu–MOF arrays exhibited a higher current density than the powder at potentials below 10 mA/cm$^2$, suggesting that the structure of the arrays can contribute to HER because of the advantage of the sponge-like structure. A comparison of the HER and OER performances with previously reported comparable electrocatalysts is summarized in Table 1. Figure 4b,c shows the EIS data measured under potential at which the current density was 10 mA/cm$^2$. The fitted results of HER (Figure 4b) revealed that the charge transfer resistance Rct of C–NiCu–MOF arrays was only 2.9 Ω cm$^2$, which was much lower than the Rct of the C–NiCu–MOF powder (9.78 Ω cm$^2$) and bare CC (21.8 Ω cm$^2$). The lower Rct indicates a faster reaction rate, suggesting that the sponge-like structure in the C–NiCu–MOF arrays can enhance the HER reaction kinetic. The fitted EIS results of OER were in the same condition, and the Rct of C–NiCu–MOF arrays was only 2.7 Ω cm$^2$, which was also smaller than Rct of the powder (8.8 Ω cm$^2$) and bare CC (50.2 Ω cm$^2$).

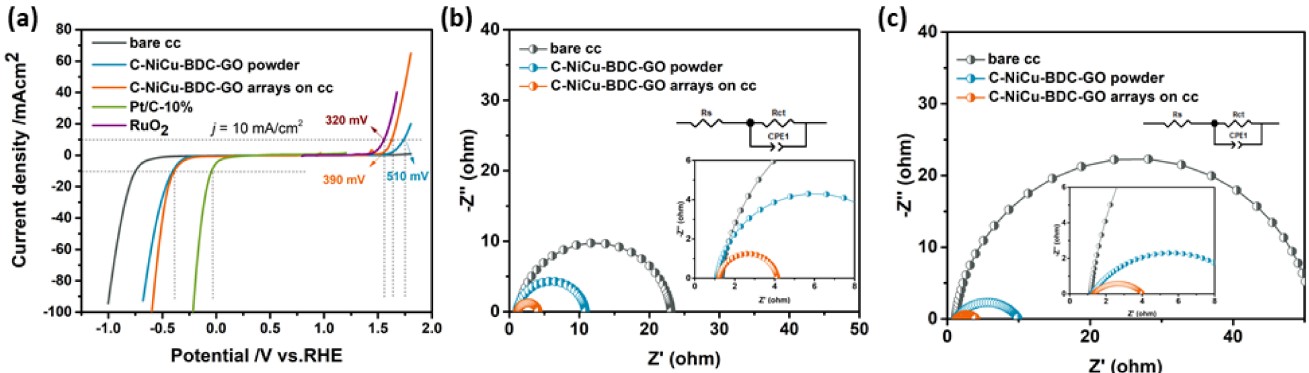

**Figure 4.** Electrochemical measurements in the three-electrode system. (**a**) LSV curves of C–NiCu–MOF arrays, powder, and bare CC. (**b,c**) Fitted EIS data of the C–NiCu–MOF arrays, powder, and bare CC for HER and OER, respectively.

**Table 1.** Comparison of the HER and OER performances with previously reported comparable electrocatalysts.

| Catalysts | Electrolyte | OER ($\eta_{10}$/mV) | HER ($\eta_{10}$/mV) | Ref. |
|---|---|---|---|---|
| Co–Fe@NC–powder | 1 M KOH | 412 | 372 | [19] |
| Co@NPC | 1 M NaOH | 360 | 325 | [21] |
| Co/NCFs | 1 M KOH | 410 | - | [36] |
| NiCo hydroxide tube | 0.1 M KOH | 460 | - | [37] |
| CoP/CDs | 1 M KOH | 400 | - | [38] |
| NiCo mixed oxide cages | 1 M KOH | 380 | - | [39] |
| NiCo mixed oxide cubes | 1 M KOH | 430 | - | [39] |
| PO–Ni/Ni–N–CNFs | 1 M KOH | 420 | - | [40] |
| HKUST–1ED | 0.5 M $H_2SO_4$ | - | 490 | [41] |
| NGO/$Ni_7S_6$ | 1 M KOH | - | 370 | [42] |
| C–NiCu–BDC MOF arrays | 1 M KOH | 390 | 400 | This work |

The overall water splitting activity of the C–NiCu–MOF arrays was conducted in the two-electrode system (Figure 5a) using two of the same 1 cm × 1 cm C–NiCu–MOF arrays as the cathode and anode simultaneously. The LSV curves (Figure 5b) were tested in 1 M KOH and revealed that the C–NiCu–MOF arrays needed the lowest potential of 2.05 V to reach a current density of 10 mA/cm$^2$, which was slightly higher than the

Pt/C‖RuO$_2$ electrode (1.8 V). It exhibited better activity compared to the potentials of the C–NiCu–MOF powder (2.17 V) and bare CC (2.75 V); besides, the C–MOF arrays exhibited a significantly higher current density than that of the C–MOF powder at the same potentials. This suggests that the structure of the arrays could contribute to the overall water splitting reaction compared with the powder. The result is consistent with that in the three-electrode system. The EIS data of the C–MOF arrays (Figure 5d), powder and bare CC were measured under potentials of 2.05, 2.1,7 and 2.75 V, respectively, at which the current density was 10 mA/cm$^2$. The fitted EIS data revealed that the charge transfer resistance Rct of the arrays was 10.3 Ω cm$^2$, which was remarkably lower than 27.4 Ω cm$^2$ of the powder. Furthermore, the EIS data for the arrays were also tested at 10 mA/cm$^2$ as well as the powder and bare CC as a comparison. The fitted result (Figure 5c) reconfirmed that the C–MOF arrays obtained the lowest Rct. Finally, the durability testing was also conducted in the same two-electrode system and evaluated at the potential of 2.05 V. The results of Figure 5c suggest that the arrays on CC maintain a stable electrocatalytic activity after 10 h of electrocatalysis, indicating that the stability of C–MOF arrays is promising. For comparison, the durability test of the C–MOF powder and bare CC were also tested at the constant current density of 10 mA/cm$^2$, as shown in Figure 5c. It clearly exhibited that arrays needed the smallest potential to reach the constant current density of 10 mA/cm$^2$ compared to the powder catalyst and bare CC. As the grey line shows in Figure 5c, the potential of bare CC at 10 mA/cm$^2$ had a continuous increase along the 10 h of electrocatalysis, which indicated that bare CC was not involved in the relevant reactions in the water splitting process. However, both the orange and blue lines showed no increase in potentials at the constant current density, which indicates that CC did not participate in the reactions because it had been modified by a layer of GO and uniform catalyst arrays had been deposited on it. These layers would prevent the direct contact between CC and the electrolyte.

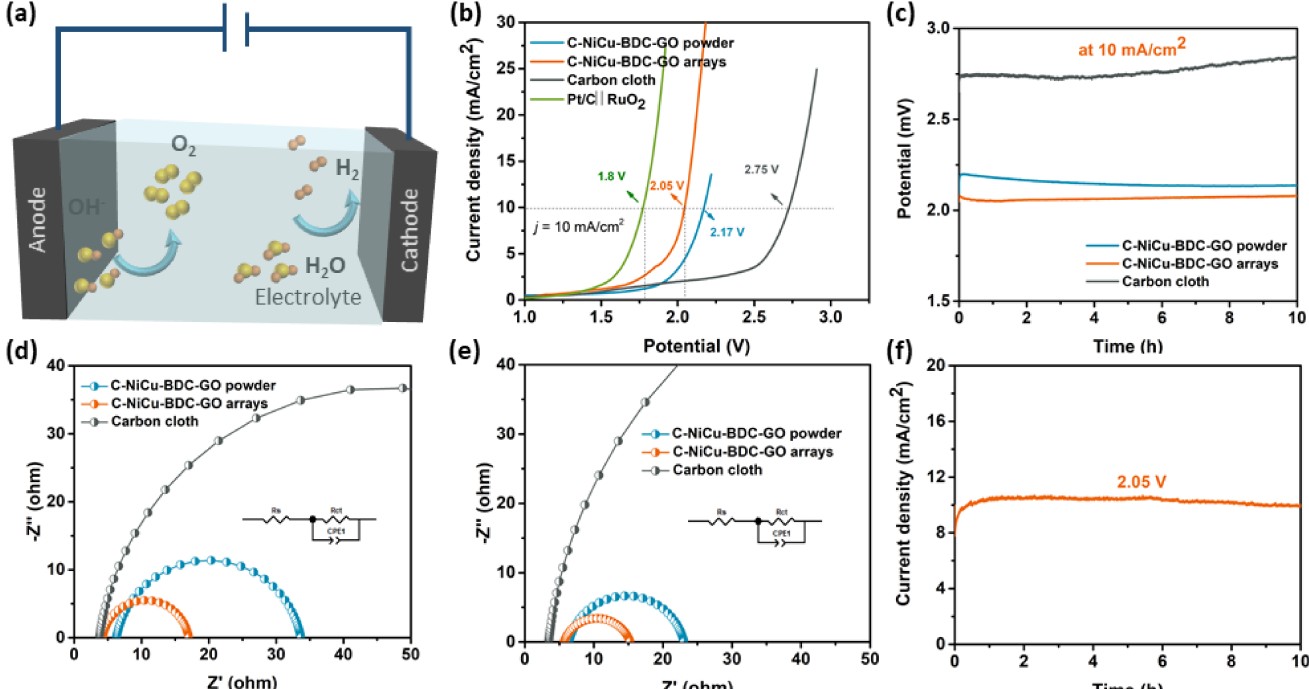

**Figure 5.** Electrochemical measurements in the two-electrode system. (**a**) Schematic illustration of the two-electrode system for overall water splitting. (**b**) LSV curves of the C–MOF arrays, powder, and CC working as both the cathode and anode electrode. (**c**) Stability test at 10 mA/cm$^2$ of the C–MOF arrays, powder, and bare CC. (**d,e**) Fitted EIS spectra of corresponding catalysts at the constant potential (the current density of the relevant potential was 10 mA/cm$^2$) and the constant current density (10 mA/cm$^2$), respectively. (**f**) Stability test of the C–MOF arrays at 2.05 V for 10 h.

## 4. Conclusions

In summary, a bimetallic MOF-derived electrocatalyst for overall water splitting was developed by a four-step procedure. MOFs were prepared by in situ transformation of NiCu–LDH arrays on GO, which can act as the anchors and the secondary conductive support. To promote the stabilities of MOF arrays, the carbonization process was involved. The characterization results showed that the MOF arrays decomposed and transferred into metal and metal oxide, not only obtaining high mechanical stability in an aqueous electrochemical system compared with pristine MOFs, but also maintaining the advantages of the structure derived from the MOF arrays. Furthermore, the electrocatalytic performance of the C–MOF arrays was remarkably better compared to its powder counterpart, which benefits from the sponge-like structure decreasing the electron transfer resistance in the electrocatalysis process of overall water splitting. This work provides a technique to modify the stability of MOFs in an aqueous system and inspires more opportunities to fabricate a bimetallic electrocatalyst.

**Supplementary Materials:** The following supporting information can be downloaded at: https://www.mdpi.com/article/10.3390/inorganics10040053/s1. Figure S1: SEM images of NiCu-LDH-GO arrays prepared at different temperatures. (a, b, c) at 180, 200, 220 °C, respectively; Figure S2: (a) 50 cycles CV of NiCu-MOFs arrays and bare CC in 0.1 M Na2SO4; (b) XRD of NiCu-MOFs arrays before and after CV testing; (c) SEM images of arrays after CV testing; Figure S3: TGA of NiCu-MOFs in N2 atmosphere; Figure S4: XRD of NiCu-LDH-GO-CC; Figure S5: (a) XPS full spectrum of NiCu-BDC-GO-CC and (b) C-NiCu-BDC-GO-CC; Figure S6: EDS of C-NiCu-BDC-GO-CC.

**Author Contributions:** Conceptualization, methodology, and validation, L.J.; P.W. and J.C.; formal analysis, investigation, and data curation, resources, writing—original draft preparation, L.J..; writing—review and editing, P.W. and J.C.; visualization, supervision, and project administration, P.W. and J.C.; funding acquisition, J.C. All authors have read and agreed to the published version of the manuscript.

**Funding:** This research was funded by the Australian Research Council Center of Excellence Scheme, CE 140100012.

**Institutional Review Board Statement:** Not Applicable.

**Informed Consent Statement:** Not Applicable.

**Data Availability Statement:** Not Applicable.

**Acknowledgments:** The authors are grateful for the financial support from the Australian Research Council Center of Excellence Scheme (CE 140100012), the Wollongong University, and ANFF. The authors would like to thank the UOW Electron Microscopy Center for the use of their equipment.

**Conflicts of Interest:** The authors declare no conflict of interest.

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
