# Peer review of "Electrocatalyst Derived from NiCu–MOF Arrays on Graphene Oxide Modified Carbon Cloth for Water Splitting"

_inorganics, doi:10.3390/inorganics10040053_

Round 1
Reviewer 1 Report
In this work, authors report a delicate method to prepare novel bimetallic metal organic framework derived electrocatalyst (C-NiCu-BDC-GO-CC), using graphene oxide (GO) modified carbon cloth
as 3D flexible and conductive substrate. The resultant electrocatalyst, CNiCu-BDC-GO-CC, exhibits very low electron transfer resistance
benefited from its extremely thin 3D sponge-like morphology.
Furthermore, it shows excellent oxygen evolution reaction (OER)
activity, achieving 10 mA/cm2 at a low overpotential of 390 mV in 1 M KOH electrolyte with remarkable durability of 10 hours. However, such overpotential is high value. Before publication in this journal authors should make following corrections:
- In figure 2 (a) XRD graph, authors should include the lattice phase of the respective peaks.
- For the overpotential calculation from the LSV, iR correction is essential. So, authors should indicate about the iR correction in the main manuscript.
- As provide LSV should compare with the benchmark electrocatalyst ( Pt/c, IrO2) for HER and OER in figure 4(a) and 5 (b).
- All the EIS figures included in this manuscript should have the equal scale in X and Y axis. Furthermore, the inset of the low Z-values should be added to show the starting point clearly.
- In figure 5 (f), Y-axis labeling is missing.
- For the data presentation style and the comparative analysis author can go through following articles with citation in the appropriate position in this manuscript: 10.1016/j.jcis.2022.03.104 , and 10.1016/j.mtnano.2021.100146 .
- From the literature survey, author should add 2 table for the comparison of such materials performance for OER and HER separately.
Author Response
Thanks reviewer for professional comments.
Please find the detail point-by-point responses in the uploaded file.

Reviewer 2 Report
I enjoyed reading this manuscript. Since a long time, there was not significant overselling in an HER/OER paper. Yet, some minor changes have to be made prior any acceptance.
1) The provided parameters in terms of current density are just reasonable for basic research. The authors should clearly state this - see e.g. https://doi.org/10.1021/jacsau.1c00092
2) In light of the aforementioned, it is questionable if the carbon support is of any relevance as carbon corrosion is on of the main issue at the anode side. This needs either to be evaluated at high current densities (above 500 mA cm-2) or at least discussed within the main manuscript.
3) The caption of Figure 5 is wrong and e and f are missing completely
4) Do the authors have any further information on the carbon corrosion of their material? Experiments to either show elevated stability or the decomposition should be provided.
Author Response
Thanks reviewer for professional comments!
Please find the detail point-by-point responses in the uploaded file.

Round 2
Reviewer 1 Report
All the comments are answered .But need some corrections.
- Pt/C -10% need to be corrected. In figures there is Pr/C written in figure 4 a and b.
- The table 1 should revised again with the overpotential values higher than this works values. (I hope authors are aware that the lesser the overpotential values efficient will be the electrocatalyst).
Reviewer 2 Report
The authors answered all my questions in a reasonable manner. As such I do have no objection towards the publication of this manuscript and recommend its acceptance.
Author Response
Thanks Reviewer!